# *Hermetia illucens* frass improves the physiological state of basil (*Ocimum basilicum* L.) and its nutritional value under drought

**Dominika Radzikowska-Kujawska**[1]*, **Zuzanna Sawinska**[1], **Monika Grzanka**[1], **Przemysław Łukasz Kowalczewski**[2], **Łukasz Sobiech**[1], **Stanisław Świtek**[1], **Grzegorz Skrzypczak**[1], **Agnieszka Drożdżyńska**[3], **Mariusz Ślachciński**[4], **Marcin Nowicki**[5]*

1 Department of Agronomy, Poznań University of Life Sciences, Poznań, Poland, 2 Department of Food Technology of Plant Origin, Poznań University of Life Sciences, Poznań, Poland, 3 Department of Biotechnology and Food Microbiology, Poznań University of Life Sciences, Poznań, Poland, 4 Institute of Chemistry and Technical Electrochemistry, Poznan University of Technology, Poznań, Poland, 5 Department of Entomology and Plant Pathology, Institute of Agriculture, University of Tennessee, Knoxville, Tennessee, United States of America

* dominika.radzikowska@up.poznan.pl (DRK); mnowicki@utk.edu (MN)

## Abstract

To counterbalance the growing human population and its increasing demands from the eco-system, and the impacts on it, new strategies are needed. Use of organic fertilizers boosted the agricultural production, but further increased the ecological burden posed by this indispensable activity. One possible solution to this conundrum is the development and application of more environmentally neutral biofertilizers. The aim of this study was to compare the effectiveness of two doses of *Hermetia illucens* frass (HI frass) with the commercial cattle manure in the cultivation of basil under drought. Soil without the addition of any organic fertilizer was used as a baseline control substrate for basil cultivation. Plants were grown with cattle manure (10 g/L of the pot volume) or HI frass at two doses (10 and 12.5 g/L). The health and physiological condition of plants were assessed based on the photosynthetic activity and the efficiency of photosystem II (chlorophyll fluorescence). Gas exchange between soil and the atmosphere were also assessed to verify the effect of fertilizer on soil condition. In addition, the mineral profile of basil and its antioxidant activity were assessed, along with the determination of the main polyphenolic compounds content. Biofertilizers improved the fresh mass yield and physiological condition of plants, both under optimal watering and drought, in comparison with the non-fertilized controls. Use of cattle manure in both water regimes resulted in a comparably lower yield and a stronger physiological response to drought. As a result, using HI frass is a superior strategy to boost output and reduce the effects of drought on basil production.

## Introduction

The consequences of climate change have formed the basis for planning food production and environmental strategies in recent years [1]. Extreme natural phenomena, such as drought and

**Data Availability Statement:** All relevant data are within the paper and its Supporting information files.

**Funding:** This work was supported by several funding sources, such as the Narodowe Centrum Badań i Rozwoju (grant no. POIR.01.01.01-00-1503/19, entitled Development of a technology for the production of organic fertilizer (in the form of pellets/granules) based on the Hermetia illucens frass and testing its impact on selected plants. Funding for open access to this research was provided by University of Tennessee's Open Publishing Support Fund. The funders had no role in study design, data collection and analysis, decision to publish, or preparation of the manuscript.

**Competing interests:** The authors have declared that no competing interests exist.

heat waves, progressively affect the condition of plants as well as the crops' yield amount and quality [2, 3]. In order to prevent further degradation of the environment, the use of various plant protection products and fertilizers is becoming increasingly more restricted [4], despite them strongly shaping the yields. Therefore, new, more ecofriendly methods of improving plant resistance to unfavorable environmental conditions are sought after, to positively affect both the condition of plants and of the environment [5–7]. One such plant-enhancing natural fertilizer can be insect frass. The use of innovative biofertilizers results from the growing number of insects bred as a complete source of animal protein for food and feed [8]. Among the insects most frequently described and cultured are the bearberry (*Tenebrio molitor*), the mealworm (*Alphitobius diaperinus*), the house cricket (*Acheta domesticus*), the black soldier fly (*Hermetia illucens*) or the house fly (*Musca domestica*) [9]. Importantly, the by-products of the agri-food industry can be used for the production of insects [10], which additionally has a positive effect on the environment and fits in with the global trends of "zero waste" and "circular economy". There are side streams during the production of insects for food and feed, including an organic fertilizer made from the excrement called frass [11]. Use of *Hermetia illucens* frass (HI frass) as an additive to the growing media in the soilless production of basil, lettuce, and tomato increased the yield (drought mass) [12]. Also, the study of the compost produced as a result of the bioconversion of coffee husks by the larvae of the *H. illucens* in the cultivation of lettuce showed a positive effect of such fertilizers on plant height, number of leaves, leaf area, and chlorophyll content [13]. Recently published data indicate that the use of insect frass as a soil component in plant cultivation provides nitrogen and other nutrients, which consequently increases plant biomass and nutrient content [11, 14]. Quality of the HI frass is high in terms of N, P, and K levels [15]. Use of HI frass from larvae increased the soil organic matter and the residual nutrients content, as well as the enzymatic activity of dehydrogenase and β-glucosidase [16].

Basil (*Ocimum basilicum* L.) belongs to Lamiaceae family and is one of the most popular herbs grown in the world., Due to its popularity around the world, basil is referred to as the "king of herbs", owing to being highly aromatic with pleasant taste used mostly in culinary [17]. Basil is also a widespread medicinal plant that is a source of essential oils [18, 19] and phenolic compounds. Phenolic acids and flavonol-glycosides are the main phenolic components in basil [20, 21]. Among the polyphenols and free phenolic acids present in basil, the most frequently described are rosmarinic, cinnamic, ferulic, vanillic, and caffeic acids [22–24]. Phenolic acids are a group of secondary metabolites with antioxidant properties, acting as reducing agents, hydrogen donors, or singlet oxygen quenchers [25]. In the context of human nutrition, phenolic acids can protect against many diseases, including cardiovascular diseases, and have anti-diabetic, anti-bacterial, anti-fungal, anti-oxidant, anti-platelet, and anti-inflammatory properties [26, 27]. The high content of phytochemicals, in particular of phenolic acids and flavonoids, renders basil a crop with substantial antioxidant activity. Basil's antioxidant activity and phenolic content is similar to raspberries and blackberries, and higher than rose hips [28]. The content of phenolic acids in basil extracts, however, largely depends on the variety, growing conditions, and extraction methods [23, 29].

Basil is an annual plant that can be successfully grown in warm and sunny conditions [30]. The optimal growth temperature ranges from 298 to 303 K [31]. Basil, however, is quite sensitive to drought stress [32]. Drought stress triggers a number of various physiological responses that have negative effects on plant growth and development. Plants are adapted to the short-term effect of the stress factor, whereas under prolonged drought stress, the rate of water absorption from the leaves is increased, which consequently reduces their hydration. This can result in closure of the stomata and in reduced cell enlargement and growth [33]. In addition, drought also causes a reduction in the activity of the electron transport chain, which leads to

the accumulation of reactive oxygen species (ROS) that are toxic at elevated levels [34]. ROS may damage nucleic acids, proteins, photosynthetic pigments, and membrane lipids [34, 35], and that damage may impair plant development, yield, and even lead to plant death.

The development of ways to improve tolerance to drought stress in plant cultivation is very useful in plant production, especially in the case of very sensitive species, including basil. Bearing the aforementioned in mind, the aim of this study was to assess whether the use of organic fertilizers, including *Hermetia illucens* frass, may have a positive effect on the physiological state of basil, as well as on the crop's mineral profile and antioxidant activity under drought.

## Materials and methods

### Materials

Light-mix soil for organic potting cultivation was purchased from Biobizz Worldwide SL (Industrial Systems s.r.o., Prague, Czech Republic), cattle manure Florovit from Grupa Inco S. A. (Góra Kalwaria, Poland) with the following composition in accordance with the manufacturer's declaration: N min. 2.8% dm, P min 2.8% dm, K min. 2.0% dm, Mg min. 0.8% dm, organic substances min. 60% dm, and soil's pH was 6.2. HI frass was purchased from HiPro-Mine S.A. (Robakowo, Poland) with the following composition in accordance with the manufacturer's declaration: N 4.23%, P 1.46%, K 3.05%, Mg 0.96%, and organic substances min. 79.3% dm. Basil seeds of the variety 'Genovese' were obtained from the company PPHU "Ogrodnik" (Poznań, Poland).

### Plant materials and growing conditions

The experiment was set up and conducted in a greenhouse belonging to the Department of Agronomy at the Poznań University of Life Science, Poznań, Poland (52.482854, 16.900465). The following greenhouse conditions were used: photoperiod 16h:8h (D:N), temperature $295.15 \pm 2$ K, sodium lighting (HPS) with a power of 400 W (Elektro-Valo Oy Netafim, Avi:13473, Uusikaupunki, Finland).

Production pots with a capacity of 1 L were filled with organic soil (Biobizz Worldwide SL), and then the analyzed fertilizers were added and mixed with the soil. Cattle manure was dosed in accordance with the manufacturer's recommendations, i.e., in the amount of 10 g/L. HI frass was applied in two doses: 10 and 12.5 g/L. I soil was then watered and basil seeds were sown at the rate of 9 seeds/pot. During plant growth, the soil was watered every 48 h in the amount of 100 mL/pot. Two parallel cultures were performed, one under the optimal constant irrigation system ('Control') and the other under drought stress ('Drought'). In order to induce the drought stress, watering was discontinued at 40 days after sowing the seeds. After 5 days of such imposed drought, during which the soil volume moisture content was monitored daily using a probe (ThetaProbe, Eijkelkamp, Netherlands), the soil moisture level of 6 to 8 vol.% was achieved. The control plants were provided with the optimum soil moisture for all analyzed variants at the level of 20 to 22 vol%. On day 46 of cultivation, the control plants and those subjected to drought stress were transferred to a phytotron where they were acclimated to darkness for 6 h to inhibit the photosynthesis. Thereafter, physiological measurements of photosynthetic activity and of chlorophyll fluorescence were performed.

### Measurements of plant photosynthetic activity

The photosynthetic activity of dark-acclimated plants was assessed using the LCpro-SD Gas Exchange Measurement System (ADC BioScientific Ltd., UK), based on the following parameters:–A—$CO_2$ assimilation level ($\mu mol\ m^{-2} s^{-1}$),–E—transpiration ($mmol\ m^{-2} s^{-1}$),–s—stomatal

conductance (mol m$^{-2}$s$^{-1}$),–i—intercellular $CO_2$ concentration (vpm). Measurements were carried out in the phytotron at a constant air temperature of 298.15 K and ambient humidity of 70±5%. The measurement sequence was the same, and the drought-stressed and non-stressed plants under the same fertilizer regimes were alternately measured. For measurements in each plant, the same, youngest, fully developed leaf was selected. The gas exchange measuring settings were set up according to the methods described earlier [36]. The concentration of $CO_2$ supplied to the measuring chamber (reference $CO_2$) was kept at 360 vpm. The flow of air supplied to the measuring chamber (u) was maintained at 200 μmol s$^{-1}$. The concentration of $H_2O$ (reference $H_2O$) was set to ambient, i.e., the actual concentration in the environment. The intensity of the light emitted in the measuring chamber (PPFD) by the red and blue (in the proportion of 10:1) LEDs of the spectrum was set to 400 μmol m$^{-2}$s$^{-1}$ (LCP Narrow Lamp, ADC BioScientific Ltd., UK). Gas exchange measurements were performed in 3 biological replications.

## Measurements of plant chlorophyll fluorescence

Chlorophyll fluorescence was measured using a Fluorometer OS5p (Optisciences Inc., Hudson, NH, USA) with the kinetic protocol selected. This allowed measuring the fluorescence of chlorophyll after dark and light adaptation and generated the following parameters: F0 –minimum fluorescence,–m—maximum fluorescence, Fv/–m—maximum photochemical efficiency of PSII, Yie–d—quantum yield of photosynthetic energy, and E–R—electron transport rate (not nominated units). The fluorescence measurement was carried out in the same way as the gas exchange measurement, the same sequence of measurements was followed and the same, youngest, fully developed leaf was selected. The Modulation Source was set to red with an intensity of 22 in the possible range from 1 to 32, where 17 corresponds to 0.1 μmol. The optimal setting is the highest possible intensity that does not induce variable fluorescence. The Saturation Flash was set to an intensity of 30 in the range of 1 to 32, with 32 being 8550 μmols. The measurement cycle was set to two saturation pulses with an interval of 180 seconds. The chlorophyll fluorescence measurements were performed in 3 biological and 2 technical replications (6 independent results).

## Fresh mass of plants

After completing the physiological measurements, the plants were cut at the soil line and weighed on a laboratory balance (RADWAG Balances and Scales, Radom, Poland).

## Soil respiration

The LCpro-SD, which was used to measure photosynthetic activity, has the ability to switch the leaf chamber for the soil respiration cylinder. A cylinder with an installed soil respiration chamber serves to close an air in one liter volume chamber order to measure the gas exchange between the soil and the atmosphere due to the activity of the biomass. The construction of the chamber consists of an acrylic dome with a built-in fan for mixing the air and a bleed-off valve preventing the formation of an excessive pressure gradient inside the chamber. The concentration of $CO_2$ supplied to the measuring chamber (reference $CO_2$), the flow of air supplied to the measuring chamber (u) and the concentration of $H_2O$ (reference $H_2O$) were set to ambient, i.e., the actual concentration in the environment. The following parameters were measured: NCER- Net $CO_2$ Exchange Rate (μmol m$^{-2}$s$^{-1}$) and W flux—Net $H_2O$ Exchange Rate (mmol m$^{-2}$s$^{-1}$). The same order of measurements was followed as in the case of the physiological measurements described above. The measurement was performed in 3 biological replicates.

## Determination of basil mineral profile

For the chemical digestion of the analytical samples, a prototype high-pressure/high-temperature system operating in a closed system with the participation of concentrated microwave energy was used. Samples of the previously freeze-dried plants (from 2 pots, in 3 repetitions, n = 6) were placed in a closed vessel with a volume of 30 mL, made of chemically modified Teflon (Hostaflon TFM). Then 3 mL of 60% nitric acid and 1 mL of 30% hydrogen peroxide were added. The vessels are placed in a steel jacket, inside which microwave energy is supplied by an antenna (power: 200 W, digestion time: 10 minutes). After mineralization, the samples were diluted to 25 mL.

In order to determine the content of elements using the ICP OES technique, the emission spectrometer with excitation source of the inductively coupled ICP plasma was used (IRIS HR, Thermo Jarell Ash, USA). The determinations were made using the calibration curve technique. The content of calcium (Ca), magnesium (Mg), potassium (K), sodium (Na), copper (Cu), iron (Fe), manganese (Mn), zinc (Zn), and lead (Pb) were analyzed. In addition, the content of phosphorus (P) was determined by the molybdate spectrophotometric method using a Specord 50 double-beam spectrophotometer (SBT) (spectral slit equal to 1.4 nm) at a wavelength of 700 nm in accordance with the method described in detail previously [37]. The results are presented in mg per g dm and were performed in 3 biological and 2 technical replications (6 independent results).

## Antioxidant activity and polyphenol profile composition

**Extraction of polyphenols.** Extraction of polyphenolic compounds was performed using 80 vol% methanol solution. The previously lyophilized sample (~1 g dm) was mixed with 15 mL of methanol solution, shaken for 30 min using a S50 laboratory shaker (CAT Germany GmbH, Lehrte, Germany) and then centrifuged at $4000 \times g$ for 20 min. The supernatant was filtered through a 0.22 μm filter and stored at -80 ˚C in a glass flask until analyzed.

**Determination of total phenolic content and antioxidant capacity.** The total content of phenolic compounds (FAE) was determined by the standard Folin-Ciocalteu colorimetric method [38] using a spectrophotometer (Multiskan GO, Thermo Fisher Scientific, Vantaa, Finland) and expressed as mg ferulic acid equivalent (FAE) per 1 g dm (mg/g dm). The total antioxidant capacity of basil were determined using the Trolox Equivalent Antioxidant Capacity (TEAC) test with the radical cation ABTS$^{\bullet+}$ according to Re at al. [39] and ferric reducing antioxidant power (FRAP) assay using the method of Benzie and Strain [40]. The antioxidant activity is expressed as TEAC value (mmol Trolox/g dm).

**Polyphenols profile composition.** The analysis of polyphenolic compounds by high performance liquid chromatography (HPLC) was performed according to the method described previously by Kowalczewski et al. [41] on the Agilent 1260 Infinity II liquid chromatograph (Agilent Technologies, Inc., Santa Clara, CA, USA) equipped with an autosampler (G7129A), a pump (G7111A), and a diode detector (G7115A) with an overview of the spectrum (190 to 400 nm). Determinations of vanillin and *p*-hydroxybenzoic acid were performed at a wavelength of 280 nm; caffeic and ferulic acids at 320 nm; whereas the chlorogenic acid at 255 nm. Phenolic compounds were separated using HPLC equipped with a SB-C18 column (50 mm x 4.6 mm with 1.8 μm particle diameter, Agilent) at 25 ˚C. The solvents used as eluents, A: water:acetic acid (98:2), B: methanol:acetic acid (98:2) at the flow of 1 mL/min, were applied in the following gradient: 0min 3% B, 3.2 min 20% B, 4.8 min 36% B, 12 min 64% B, 13 min 100% B, 16 min 100% B, 17 min 3% B, 25 min 3% B. The samples were applied to the column in the amount of 6 mL. Quantitative calculations were made using peak areas by measurement and computer integration using OpenLab CDS (Agilent Technologies, Inc., Santa Clara, CA, USA) and the results were expressed as μg/g dm.

## Statistical analysis

Statistical analysis of the data was performed using Statistica 13 (Dell Software Inc., USA) software and R v4.1.2 [42] with package *agricolae* v1.3–5 [43]. For every test, three independent measurements were taken, unless stated otherwise. All measurements were studied using one-way analysis of variance independently for each dependent variable, or two-way ANOVA for the assessment of factorial interactions. Post-hoc Tukey honest significant difference (HSD) multiple comparison tests were used to identify statistically homogeneous subsets at α = 0.05.

## Results and discussion

### Measurements of plant photosynthetic activity

Fertilizers are an effective means of plant reinforcement under stresses, especially of the drought in many plants, including basil. In this study, we examined whether the physiological and biochemical changes due to biofertilizers application able to increase the tolerance of basil plants to mild water stress, a condition that repeats itself during each growing season. The assessment of photosynthetic activity was carried out on the basis of the measurement of gas exchange: $CO_2$ assimilation (A), $H_2O$ transpiration (E), stomatal conductance (Gs) and intercellular $CO_2$ concentration (Ci) (Table 1). Factorial single- and two-way ANOVA analysis provided information on the significant impacts of the water regime, the fertilization conditions, and the combination thereof (S1 Table). Overall, drought impacts proved significant at lower *p* than fertilization regimes, and the combination of both factors was significant for all parameters but E. The highest level of A, and thus the greatest efficiency of photosynthesis, both under optimal conditions and under drought, was recorded for plants treated with HI frass in both doses. Under drought, it exceeded the respective values in the non-stressed plants by 205% (10g/L) and by 212% (12.5 g/L). The use of cattle manure also significantly improved A compared to the non-fertilized plants under optimal conditions (by 36%) or drought (by 155%), respectively. The beneficial effects of fertilization with manure on drought tolerance were previously observed in wheat [44]. Significantly higher results of $CO_2$ assimilation under stress were noted for plants fertilized with manure. In that same study, a significant improvement under optimal conditions was also attributed to fertilization [44]. Here, we observed no significant differences in the photosynthetic activity between plants growing under optimal conditions and drought among the fertilized basil. A comparably larger significant difference was observed for the non-fertilized plants under both water regimes. As such, our data provide another proof for the positive effect of the fertilizer use on the photosynthetic activity. Those results also prove that in the fertilized basil, the imposed drought lacked a strong intensity and failed to significantly reduce the photosynthetic activity. Our data imply only partial stomatal closure, which allowed to maintain A and E under drought. There were no significant differences in E or Ci between the fertilized basil under optimal conditions and under drought, but these parameters differed significantly in the non-fertilized plants.

Notably, a significant decrease in Gs as a result of drought was noted in both non-fertilized and fertilized plants. Significantly higher values of Gs were observed in plants fertilized with both doses of HI frass under both water regimes (Table 1). HI frass use improved Gs in drought stressed basil six- or five-fold, respectively. The observed small reduction of A is because of limited Gs that regulate practically concurrently A and E [45]. Comparably, in wheat the reduction of E and Gs due to drought was the strongest in plants fertilized with manure [44], which was attributed to defense mechanisms against damage to the photosynthetic apparatus, but this was not noted in our study of basil. The level of imposed drought was likely moderate, and the use of fertilizers, especially HI frass, seemingly prevented these drought protective mechanisms

**Table 1. Parameters of basil photosynthetic activity (columns) at various fertilizer formulations (rows), under optimal conditions (control) and under drought (drought).**

| Fertilizer | Dose of fertilizer (g/L) | A | | | E | | | Gs | | | Ci | | |
|---|---|---|---|---|---|---|---|---|---|---|---|---|---|
| | | control | drought | Mean[1A] | control | drought | Mean[1A] | control | drought | Mean[1A] | control | drought | Mean[1A] |
| Control | - | $6.50 \pm 0.42^c$ | $3.37 \pm 0.10^d$ | 4.94c | $2.32 \pm 0.69^{abc}$ | $0.61 \pm 0.01^d$ | 1.47b | $0.14 \pm 0.01^c$ | $0.02 \pm 0.01^e$ | 0.080c | $265 \pm 25^a$ | $163 \pm 4^e$ | 214.33a |
| Cattle manure | 10 | $8.82 \pm 0.17^b$ | $8.59 \pm 0.71^b$ | 8.70b | $2.43 \pm 0.11^{abc}$ | $1.48 \pm 0.08^{cd}$ | 1.96ab | $0.14 \pm 0.01^c$ | $0.08 \pm 0.01^d$ | 0.108bc | $245 \pm 10^{ab}$ | $190 \pm 3^{de}$ | 217.50a |
| HI frass | 10 | $10.76 \pm 0.38^a$ | $10.28 \pm 0.40^a$ | 10.52a | $3.057 \pm 0.18^a$ | $2.45 \pm 0.07^{abc}$ | 2.75a | $0.20 \pm 0.01^a$ | $0.15 \pm 0.01^{bc}$ | 0.175a | $258 \pm 3^{ab}$ | $207 \pm 8^{cd}$ | 232.83a |
| HI frass | 12.5 | $11.18 \pm 0.68^a$ | $10.53 \pm 0.45^a$ | 10.86a | $2.67 \pm 0.90^{ab}$ | $1.74 \pm 0.13^{bcd}$ | 2.21ab | $0.17 \pm 0.02^{ab}$ | $0.13 \pm 0.01^c$ | 0.153ab | $229 \pm 24^{bc}$ | $188 \pm 15^{de}$ | 208.50a |
| Mean[1A] | | 9.31a | 8.19a | | 2.62a | 1.57b | | 0.162a | 0.097b | | 249.42a | 187.17b | |

Different letters a–e indicate statistically different mean values ($\alpha = 0.05$). $CO_2$ assimilation level–A ($\mu$mol m$^{-2}$ s$^{-1}$), $H_2O$ transpiration–E (mmol m$^{-2}$ s$^{-1}$), stomatal conductance–Gs (mol m$^{-2}$ s$^{-1}$) and intercellular $CO_2$ concentration–Ci ($\mu$mol mol$^{-1}$) in drought stress depending on the fertilizer used.

[1A]: Results of independent 1-way ANOVA for either factor (Fertilizer; water stress). A: LSD$_{Stress}$: 2.17; LSD$_{Fertilizer}$: 1.58; LSD$_{Stress*Fertilizer}$: 1.31. E: LSD$_{Stress}$: 0.54; LSD$_{Fertilizer}$: 1.16; LSD$_{Stress*Fertilizer}$: 1.17. Gs: LSD$_{Stress}$: 0.036; LSD$_{Fertilizer}$: 0.064; LSD$_{Stress*Fertilizer}$: 0.025. Ci: LSD$_{Stress}$: 16.73; LSD$_{Fertilizer}$: 62.48; LSD$_{Stress*Fertilizer}$: 40.81.

from being triggered. In the research on the effect of nano-organic fertilizers on tomato plants under drought, results similar to ours were noted [46]. The applied fertilizers allowed to maintain both A and E in tomato plants under drought at levels exceeding those in non-fertilized plants. In the study of the effect of natural fertilizers on tomato growth, the fertileizers had a positive effect on the physiological condition of plants and effectively increased the efficiency of photosynthesis [47]. Results of the fertilized date palms showed an increase in leaf water potential, Gs, Fv/Fm, and chlorophyll pigment synthesis [48]. All those convergent data point to the fact that organic fertilizers, owing to the nutrients they contain, have the ability to improve soil texture, plant nutrition, and soil water retention.

## Measurements of plant chlorophyll fluorescence

The parameters of chlorophyll fluorescence are subject to various modifications depending on the type, duration and intensity of stress. These parameters make it possible to detect changes in PSII functioning that result from the stress experienced by the plants before any visible symptoms of damage manifest themselves [49]. Factorial single- and two-way ANOVA analysis provided information on the significant impacts of the water regime, fertilization conditions, and combination thereof (S1 Table). Overall, fertilization impacts proved significant at $p$ comparable to water regime, and the combination of factors was significant for all parameters but Fv/Fm. Based on the analysis of the results of chlorophyll fluorescence (Fig 1), we found a significant positive effect of the application of both doses of HI frass on the following parameters measured after the plants acclimated to darkness: minimum fluorescence (F0), maximum fluorescence (Fm), maximum photochemical efficiency of PSII (Fv/Fm), as well as on the parameters measured in the light: quantum yield of photosynthetic energy (Yield) and electron transport rate (ETR), under optimal conditions and under drought. Significant differences in the values F0 under drought were observed. The most favorable, i.e., the lowest values of F0 were recorded for plants fertilized with HI frass 12.5 g/L, followed by HI frass 10 g/L, and then cattle manure, and the worst–for the non-fertilized plants.

The F0 of dark-adapted leaf is an indicator of the excitation energy loss during its transfer to the PSII reaction center [50]. Higher values of that parameter due to drought in the non-fertilized plants may indicate their lower efficiency of the excitation energy transfer between photosynthetic complexes. Significantly the highest values of Fm have been observed in basil fertilized with HI frass 10g/L, both under optimal hydration conditions and under drought stress. Reduction of the value of Fm indicates the occurrence of stress, as a result of which not all electron acceptors in PSII were completely reduced [51]. Similarly, the Fv/Fm was significantly the highest in drought stressed basil plants fertilized with HI frass 10g/L, however, in all fertilized plants Fv/Fm was significantly higher than in their non-fertilized controls. Similar to a study of wheat that also used cattle manure [44], the natural fertilizers had a positive effect on the functioning of PSII under drought in our study of basil. Among the plants growing under the optimal conditions, significantly lower values of this parameter were observed only in the non-fertilized basil (Fig 1). A beneficial effect of cattle manure on Fv/Fm was also noted for wheat under optimal hydration [44]. The same results were obtained by for tomato plants [46], wherein the use of nano-organic fertilizers (nano-vermicompost) significantly improved Fv/Fm both under optimal hydration and under drought. The Fv/Fm values ranging from 0.7 to 0.8 are typical for most plant species in the absence of stress factors [52]; thus, drought intensity imposed by us on the basil plants did not cause permanent damage to PSII. Non-photochemical quenching (NPQ) dissipates the excitation energy and quenches the chlorophyll fluorescence. Plants are able to maintain a low chlorophyll fluorescence due to NPQ, which helps minimizing the generation of singlet oxygen in the PSII antenna [53]. A decrease in

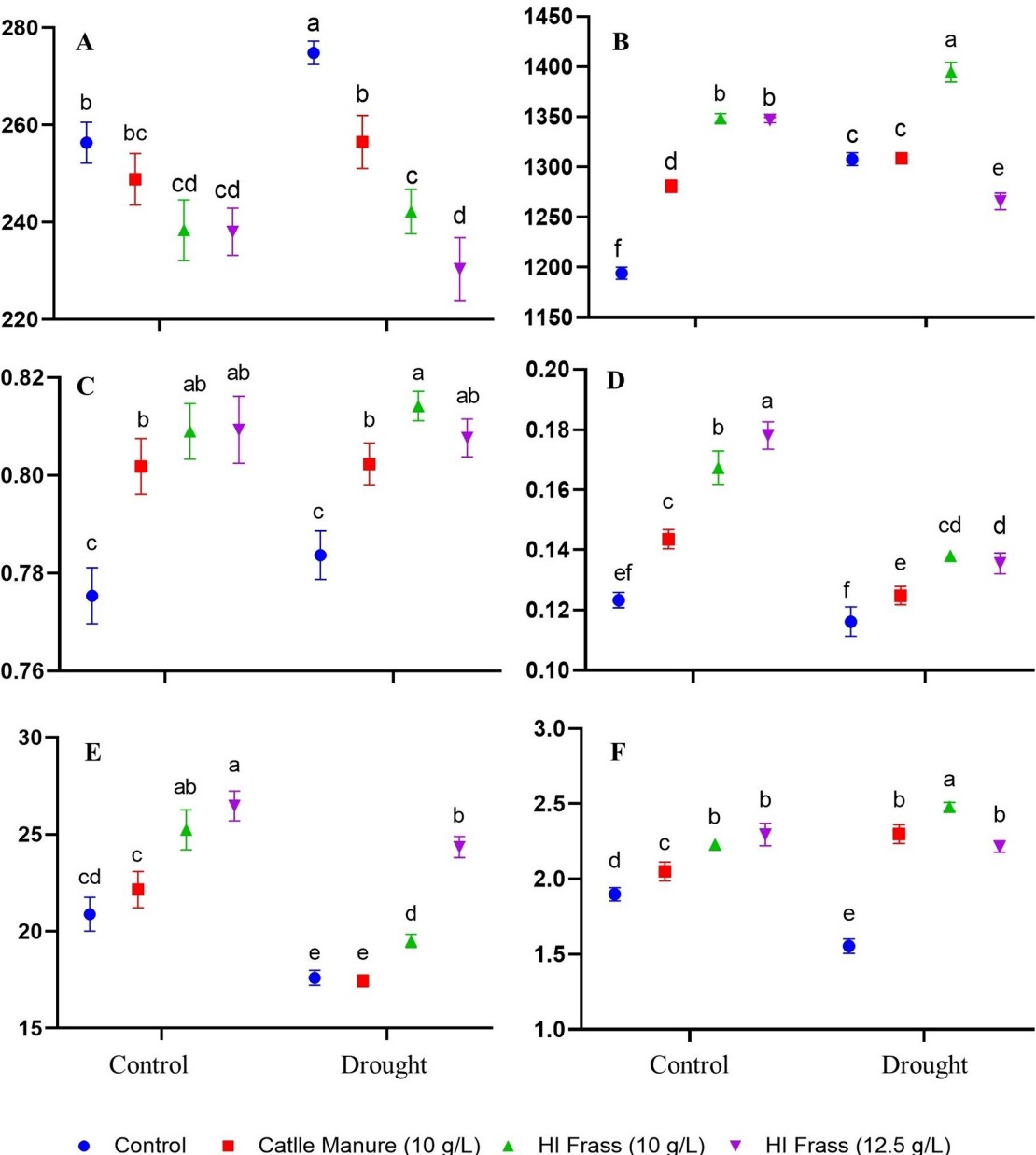

**Fig 1. Parameters of chlorophyll fluorescence of the basil plants cultivated under two water regimes and using various fertilizer formulations.** Parameters assessed after dark adaptation: A- minimum fluorescence (F0), B- maximum fluorescence (Fm), C- maximum photochemical efficiency of PSII (Fv/Fm). Parameters assess in the light: D- quantum yield of photosynthetic energy (Yield), E- Electron Transport Rate (ETR) and F- non-photochemical quenching (NPQ) [non-nominated units]. Letters a-f indicate statistically different mean values ($\alpha = 0.05$). F0: $LSD_{Stress}$: 8.16; $LSD_{Fertilizer}$: 8.15; $LSD_{Stress*Fertilizer}$: 9.38. Fm: $LSD_{Stress}$: 33.16; $LSD_{Fertilizer}$: 46.06; $LSD_{Stress*Fertilizer}$: 11.62. Fv/Fm: $LSD_{Stress}$: 0.008; $LSD_{Fertilizer}$: 0.006; $LSD_{Stress*Fertilizer}$: 0.0095. Yield: $LSD_{Stress}$: 0.0098; $LSD_{Fertilizer}$: 0.0163; $LSD_{Stress*Fertilizer}$: 0.007. ETR: $LSD_{Stress}$: 1.55; $LSD_{Fertilizer}$: 2.49; $LSD_{Stress*Fertilizer}$: 1.29. NPQ: $LSD_{Stress}$: 0.163; $LSD_{Fertilizer}$: 0.152; $LSD_{Stress*Fertilizer}$: 0.095.

NPQ as a result of drought was recorded only in the non-fertilized basil plants. In plants fertilized with manure and HI frass at a dose of 10 g/L, an increase in the NPQ value was observed, whereas in those fertilized with HI frass at the higher dose, no significant differences vs, non-fertilized controls were found.

The significant influence of the applied fertilizers on Yield and ETR both under drought stress and control conditions, has been observed. The greatest effect was noted for both doses of HI frass; these treatments improved Yield under drought by 17%, whereas ETR by 11% and 39%, respectively. Drought significantly decreased the values of both Yield and ETR (Fig 1). We also note a decrease in the value of both of these parameters due to drought in our previous studies of other plants [36, 49, 54]. Photosynthesis efficiency in barley [55] showed only a slight reduction in A, E, and Gs due to drought, which likely reduced the demand for ATP and NADPH, and also reduced Yield and ETR. Fertilized basil plants, especially those fertilized with HI frass, maintain (partially) open stomata, high $CO_2$ supply, and ample $CO_2$ assimilation; all this suggests an effective electron sink, which must be accompanied by high Yield.

## Fresh mass of plants

Drought stress caused a decrease of basil fresh mass in the above-ground part for all fertilizers used in the experiment. This fresh mass loss ranged from 50% for cattle manure to 54% for HI frass at a dose of 10 g/L (Table 2). Similar decreases were observed upon application of nano-fertilizer of paulownia at the different irrigation levels on basil dry weight [56]. In that same study, an increase in dry matter of roots under drought was noted [56], an observation not found in our study. Another study on drought impacts on the basil plants also noted significant decreases in fresh and dry mass of leaf and roots [57]. In that study, pollen grains extract used as fertilizer under drought caused an increase in fresh weight of the above-ground part and roots by 44 and 38%, respectively [57]. Factorial single- and two-way ANOVA analysis provided information on the significant impacts of the water regime, fertilization conditions, and combination thereof (S1 Table). Overall, drought impacts proved significant at lower $p$ than fertilization, and the combination of factors was not significant for all parameters. Comparably, no significant impacts on the values of fresh mass were found between the tested fertilizers under both tested water regimes; however, the plants treated with HI frass at a dose of 10 g/L had apparently the highest fresh mass. Notably, under optimal hydration, the use of HI frass at a dose of 10 g/L resulted in a significant increase of 58% in fresh mass of whole plants compared to non-fertilized plants (Table 2).

## Soil respiration

The growing popularity of biofertilizers aligns well with the environmental protection, as they bring many benefits in the agricultural systems of plant production. Appropriate use of

**Table 2. Fresh mass (FM) of basil roots, above-ground part, and whole plants (g) under drought stress depending on the fertilizer used.**

| Fertilizer | Dose of fertilizer (g/L) | FM of roots | | | FM of above-ground part | | | FM of whole plants* | | |
|---|---|---|---|---|---|---|---|---|---|---|
| | | control | drought | Mean[1A] | control | drought | Mean[1A] | control | drought | Mean[1A] |
| Control | - | 0.82 ± 0.21[a] | 1.08 ± 0.06[a] | 0.95ab | 15.68 ± 3.73[bc] | 8.33 ± 1.50[c] | 12.01a | 16.5 [bcd] | 9.4 [d] | 12.96a |
| Cattle manure | 10 | 1.14 ± 0.37[a] | 0.57 ± 0.05[a] | 0.85ab | 19.98 ± 5.82[ab] | 9.99 ± 2.52[c] | 14.99a | 21.1 [ab] | 10.6 [d] | 15.84a |
| HI frass | 10 | 1.61 ± 0.83[a] | 1.52 ± 0.55[a] | 1.57a | 24.52 ± 4.26[a] | 11.37 ± 2.24[c] | 17.95a | 26.1 [a] | 12.9 [cd] | 19.51a |
| HI frass | 12.5 | 1.04 ± 0.54[a] | 0.98 ± 0.18[a] | 1.01ab | 18.85 ± 9.01[ab] | 9.18 ± 0.83[c] | 14.01a | 19.9 [abc] | 10.2[d] | 15.02a |
| Mean[1A] | | 1.15a | 1.04a | | 19.76a | 9.72b | | 20.91a | 10.76b | |

*calculated as a sum of roots and above-ground parts of plants. Different letters a-d indicate statistically different mean values (α = 0.05).

[1A]: Results of independent 1-way ANOVA for either factor (Fertilizer; water stress). FM of roots: LSDStress: 0.42; LSDFertilizer: 0.69; LSDStress*Fertilizer: 1.23. FM of above ground parts: LSDStress: 3.87; LSDFertilizer: 11.16; LSDStress*Fertilizer: 12.70. FM of whole plants: LSDStress: 4.20; LSDFertilizer: 11.56; LSDStress*Fertilizer: 12.70.

**Table 3. Soil respiration in basil cultivated under drought stress, depending on the fertilizer used.**

| Fertilizer | Dose of fertilizer (g/L) | W flux | | | NCER | | |
|---|---|---|---|---|---|---|---|
| | | control | drought | Mean[1A] | control | drought | Mean[1A] |
| Control | - | $80.5 \pm 13.21^b$ | $24.2 \pm 2.38^c$ | 52.31c | $8.47 \pm 1.39^b$ | $2.55 \pm 0.25^c$ | 5.5 |
| Cattle manure | 10 | $77.7 \pm 6.16^b$ | $29.4 \pm 1.38^c$ | 53.53c | $8.17 \pm 0.65^b$ | $3.10 \pm 0.15^c$ | 5.64c |
| HI frass | 10 | $89.5 \pm 16.39^b$ | $55.1 \pm 3.91^{bc}$ | 72.30b | $9.42 \pm 1.73^b$ | $5.80 \pm 0.41^{bc}$ | 7.61b |
| HI frass | 12.5 | $150.9 \pm 41.27^a$ | $81.1 \pm 2.22^b$ | 115.97a | $15.88 \pm 4.34^a$ | $8.53 \pm 0.23^b$ | 12.21a |
| **Mean[1A]** | | 99.62a | 47.44b | | 10.49a | 4.99b | |

Different letters a-d indicate statistically different mean values ($\alpha = 0.05$). W flux—Net $H_2O$ Exchange Rate; NCER-Net $CO_2$ Exchange Rate.

[1A]: Results of independent 1-way ANOVA for either factor (Fertilizer; water stress). W flux: $LSD_{Stress}$: 9.81; $LSD_{Fertilizer}$: 16.80; $LSD_{Stress*Fertilizer}$: 14.72. NCER: $LSD_{Stress}$: 0.94; $LSD_{Fertilizer}$: 1.77; $LSD_{Stress*Fertilizer}$: 1.55

biofertilizers improves the quality of the soil and increases the availability of water, and provides an economically attractive source of nutrients for plants [58]. Numerous studies proved that the use of manure, by increasing the biodiversity of soil organisms, among others effects, increases the soil respiration [59–61]. Contrastingly, drought stress reduces both the C uptake and soil respiration [62, 63]. Factorial single- and two-way ANOVA analysis provided information on the significant impacts of the water regime, fertilization conditions, and combination thereof (S1 Table). Overall, drought impacts proved significant at *p* comparable to fertilization, and the combination of factors was significant for all studied parameters. Our data also indicate that drought stress causes a reduction of NCER and a reduction of W flux in all tested fertilizers and in non-fertilized plants, alike (Table 3). Notably, significantly the highest soil respiration, both under optimal conditions and under drought, was recorded in plants fertilized with HI frass at the dose of 12.5 g/L followed by HI frass at 10 g/L (Table 3). The decrease in both $CO_2$ and $H_2O$ exchange rates due to imposed drought was found to be significantly the lowest in plants fertilized with HI frass at 10 g/L (38%), followed by HI frass at 12.5 g/L (46%), and in plants fertilized with cattle manure (62%). In the non-fertilized plants, the decrease in soil respiration due to drought imposition was 70% (Table 3).

## Determination of basil mineral profile

Minerals, such as Ca, Mg, Cu, Zn, or Fe, are important nutrients for normal growth and function of the human body [64]. Basil is a source of many minerals, including Fe, Mn, Zn, Mg, Ca, P, K, Na [65]. Both the use of (bio)fertilizers, as well as abiotic stresses such as high salt levels, drought, or extreme temperatures, can substantially affect the mineral profile of basil that is then consumed by humans [66]. As the abiotic stresses become increasingly more important due to the global water scarcity, they may directly contribute to changes in the nutritional quality of food [67]. This prompted our analyses of the mineral profiles of basil cultivated under various conditions (Table 4).

Under both optimal hydration and drought alike, the use of HI frass resulted in the accumulation of a greater amount of Mg in basil biomass. This is a beneficial phenomenon from the nutritional point of view [68]. Magnesium in the plant is also an essential component of chlorophyll, so we posit that its amount is also significantly higher, and the plants are more colored (green). This was confirmed by the results of chlorophyll fluorescence analysis (see section 3.2.), which indicated a significantly higher activity of chlorophyll in plants fertilized with HI frass. Due to fertilization with HI frass, the content of K -an activator of many

Table 4. The content of minerals in basil cultivated under various growing conditions and using various fertilizer formulations.

| Watering condition | Fertilizer | Dose of fertilizer (g/L) | Ca | Mg | K | Na | Cu | Fe | Mn | Zn | Pb | P |
|---|---|---|---|---|---|---|---|---|---|---|---|---|
| Control | Control | - | $138 \pm 11^a$ | $23.7 \pm 2.5^b$ | $39.9 \pm 2.8^b$ | $2.55 \pm 0.18^a$ | $0.21 \pm 0.02^a$ | $0.09 \pm 0.01^c$ | $0.19 \pm 0.01^c$ | $0.08 \pm 0.01^b$ | $0.44 \pm 0.04^a$ | $28.2 \pm 2.0^b$ |
| | Cattle manure | 10 | $114 \pm 10^b$ | $39.5 \pm 2.8^a$ | $35.5 \pm 2.6^{ab}$ | $3.00 \pm 0.18^a$ | $0.24 \pm 0.02^a$ | $0.09 \pm 0.01^c$ | $0.23 \pm 0.01^b$ | $0.08 \pm 0.01^b$ | $0.49 \pm 0.03^a$ | $27.4 \pm 1.9^b$ |
| | HI frass | 10 | $107 \pm 9^b$ | $41.7 \pm 3.0^a$ | $45.2 \pm 3.2^a$ | $12.5 \pm 1.4^b$ | $0.22 \pm 0.04^a$ | $0.22 \pm 0.01^b$ | $0.30 \pm 0.02^a$ | $0.11 \pm 0.01^a$ | $0.46 \pm 0.02^a$ | $27.4 \pm 1.9^b$ |
| | HI frass | 12.5 | $101 \pm 9^b$ | $38.0 \pm 2.7^a$ | $41.0 \pm 3.0^a$ | $18.4 \pm 1.7^b$ | $0.10 \pm 0.01^b$ | $1.53 \pm 0.08^a$ | $0.19 \pm 0.01^c$ | $0.05 \pm 0.01^c$ | $0.46 \pm 0.01^a$ | $34.1 \pm 1.7^a$ |
| Drought | Control | - | $138 \pm 11^A$ | $39.9 \pm 2.8^A$ | $37.6 \pm 2.8^B$ | $2.11 \pm 0.15^B$ | $0.22 \pm 0.03^A$ | $0.11 \pm 0.01^B$ | $0.30 \pm 0.02^A$ | $0.06 \pm 0.01^A$ | $0.45 \pm 0.02^{AB}$ | $20.7 \pm 1.4^{AB}$ |
| | Cattle manure | 10 | $107 \pm 9^B$ | $34.1 \pm 2.9^B$ | $37.7 \pm 2.5^B$ | $1.85 \pm 0.14^C$ | $0.23 \pm 0.02^A$ | $0.09 \pm 0.01^C$ | $0.18 \pm 0.01^B$ | $0.05 \pm 0.01^A$ | $0.48 \pm 0.03^A$ | $19.9 \pm 1.5^B$ |
| | HI frass | 10 | $100 \pm 8^B$ | $37.3 \pm 2.9^{AB}$ | $37.8 \pm 2.7^B$ | $1.83 \pm 0.14^C$ | $0.20 \pm 0.06^A$ | $0.19 \pm 0.01^A$ | $0.19 \pm 0.01^B$ | $0.04 \pm 0.01^A$ | $0.40 \pm 0.02^B$ | $18.3 \pm 1.4^B$ |
| | HI frass | 12.5 | $105 \pm 10^B$ | $43.1 \pm 3.0^A$ | $46.6 \pm 3.7^A$ | $2.50 \pm 0.18^A$ | $0.19 \pm 0.04^A$ | $0.21 \pm 0.02^A$ | $0.30 \pm 0.02^A$ | $0.06 \pm 0.01^A$ | $0.36 \pm 0.02^B$ | $22.5 \pm 1.7^A$ |

The results noted with different letters (lowercase for optimal watering conditions, uppercase for drought stress) differ statistically significantly at the level of $\alpha = 0.05$.

enzymes and an element necessary to the maintenance of cell membrane potential [69]- also increased significantly. In basil grown under optimal conditions, the content of P was comparably the highest. Drought limited the absorption of P from the soil and no beneficial effects of biofertilizers on its content were observed. HI frass fertilization, on the other hand, caused a significant increase in Fe content. Moreover, the increase in Fe content was dependent on the dose applied: The higher the dose, the greater the observed increase in the content of Fe in the basil. Changes in the observed content of key minerals are consistent with the findings of other scholars, who showed an increase in the content of important minerals in basil due to vermicompost fertilization [70]. Our use of biofertilizers lowered the Ca content in all tested plants, grown both under optimal conditions and drought. Notably, we recorded significantly lower accumulation of Pb in basil fertilized with HI frass than with the cattle manure.

## Antioxidant activity and polyphenol profile composition

The total phenolic content of basil depends on environmental conditions during growth, including temperature, photoperiod, soil nutrient availability, as well as seasonal, geographic, and climatic variability [71]. When exposed to unfavorable environmental conditions, plants generate more free radicals. In order to ensure the oxidative balance, metabolites with antioxidant properties are synthesized, to protect plants against the degenerative changes [72]. Our results show that the content of phenolic compounds (FAE) in basil methanolic extracts significantly differed depending on the type of fertilization used (Table 5). The use of HI frass reduces the drought responses in basil. Under optimal conditions, FAE ranged from 18.2 mg/g for plants fertilized by HI frass at 12.5 g/L to 25.1 mg/g for plants fertilized by cattle manure, whereas under drought from 25.9 mg/g to 35.2 mg/g, respectively. The FAE values for plants under optimal conditions are consistent with the literature data: the content of phenolic compounds in 15 varieties of basil ranged from 3.47 to 17.58 mg/g [23]. The higher the fertilizer dose, the less polyphenolic compounds and the lower the antioxidant activity (TEAC value). Again, as for FAE, the use of HI frass reduced the TEAC measured by either method. This proves a beneficial effect of fertilization on the plants experiencing changes in the environmental conditions, and in the case of basil subjected to drought, an improved tolerance to this stress. Cattle manure, here considered a reference fertilizer, caused an increase in the content of polyphenols and antioxidant activity, likely due to an inherent, additional osmotic stress effect. Other related studies also indicated differences in the antioxidant activity of basil depending on the fertilization used [73, 74].

**Table 5. Minerals composition of basil cultivated under various water regimes and under various fertilizer formulations.**

| Fertilizer | Dose of fertilizer (g/L) | FAE | | | TEAC$_{ABTS}$ | | | TEAC$_{FRAP}$ | | |
|---|---|---|---|---|---|---|---|---|---|---|
| | | Control | Drought | Mean[1A] | Control | Drought | Mean[1A] | Control | Drought | Mean[1A] |
| Control | - | 23.3 ± 1.6[b] | 28.4 ± 1.4[BC] | 25.89a | 195 ± 11[b] | 165 ± 11[B] | 180.33ab | 27.81 ± 0.47[a] | 35.16 ± 2.83[A] | 29.95a |
| Cattle manure | 10 | 25.1 ± 2.5[a] | 35.2 ± 2.2[A] | 25.56a | 208 ± 13[a] | 186 ± 13[A] | 197.17a | 26.06 ± 2.12[a] | 35.80 ± 4.01[A] | 29.87a |
| HI frass | 10 | 23.1 ± 1.4[b] | 29.2 ± 3.3[B] | 26.21a | 188 ± 18[b] | 134 ± 16[C] | 162.00bc | 24.73 ± 2.06[ab] | 34.15 ± 3.86[A] | 31.81a |
| HI frass | 12.5 | 18.2 ± 1.7[c] | 25.9 ± 25[C] | 26.75a | 133 ± 16[c] | 128 ± 12[C] | 129.33c | 22.28 ± 1.98[b] | 33.67 ± 4.02[A] | 28.22a |
| **Mean[1A]** | | 22.48a | 29.72b | | 180.83a | 153.58b | | 25.22a | 34.70b | |

The results noted with different letters (lowercase for optimal watering conditions, uppercase for drought stress) differ statistically significantly at the level of α = 0.05. FAE–ferulic acid equivalent; TEAC$_{ABTS}$–Trolox-equivalent antioxidant capacity measured by ABTS method; TEAC$_{FRAP}$–Trolox-equivalent antioxidant capacity measured by FRAP method.

[1A]: Results of independent 1-way ANOVA for either factor (Fertilizer; water stress). FAE: LSD$_{Stress}$: 3.79; LSD$_{Fertilizer}$: 9.91; LSD$_{Stress*Fertilizer}$: 10.55. TEAC$_{ABTS}$: LSD$_{Stress}$: 25.66; LSD$_{Fertilizer}$: 35.16; LSD$_{Stress*Fertilizer}$: 39.27. TEAC$_{FRAP}$: LSD$_{Stress}$: 2.46; LSD$_{Fertilizer}$: 9.46; LSD$_{Stress*Fertilizer}$: 7.99

**Table 6. Phenolic compounds determined by HPLC in basil methanolic extracts, from plants cultivated under various water regimes and with various fertilizer formulations.**

| Watering condition | Fertilizer | Dose of fertilizer (g/L) | *p*-Hydroxybenzoic acid (µg/g dm) | | Chlorogenic / caffeic acids* (µg/g dm) | | Vanillin (µg/g dm) | | Ferulic acid (µg/g dm) | |
|---|---|---|---|---|---|---|---|---|---|---|
| | | | **Control** | **Drought** | **Control** | **Drought** | **Control** | **Drought** | **Control** | **Drought** |
| Control | Control | - | 1.463 ± 0.062[b] | 0.036 ± 0.009[D] | 107 ± 6.6[a] | 134 ± 8.1[A] | 0.206 ± 0. 13[a] | 0.926 ± 0.038[D] | 735 ± 33[c] | 1400 ± 71[B] |
| | Cattle manure | 10 | 1.371 ± 0.048[b] | 0.966 ± 0.032[A] | 89.4 ± 4.9[b] | 122 ± 4.2[B] | 0.065 ± 0.003[b] | 1.943 ± 0.050[A] | 993 ± 56[a] | 1523 ± 66[A] |
| | HI frass | 10 | 0.540 ± 0.033[c] | 0.456 ± 0.028[B] | 74.1 ± 4.7[c] | 128 ± 6.7[B] | N/D | 1.601 ± 0.047[B] | 867 ± 48[b] | 1511 ± 59[A] |
| | HI frass | 12.5 | 2.010 ± 0.021[a] | 0.303 ± 0.033[C] | 87.3 ± 8.8[bc] | 88.7 ± 6.2[C] | 0.181 ± 0.014[a] | 1.284 ± 0.022[C] | 836 ± 55[b] | 1559 ± 92[A] |

* determined and calculated based on an external chlorogenic acid standard. N/D–not detected. The results noted with different letters (lowercase for optimal watering conditions, uppercase for drought stress) differ statistically significantly at the level of $\alpha = 0.05$

Literature data indicate that basil is a source of many polyphenolic compounds [22–24]. Here, we determined the content of ferulic, *p*-hydroxybenzoic, chlorogenic, caffeic acids, and vanillin in the analyzed extracts (Table 6). Importantly, basil grown under various conditions and fertilized with various formulations showed different profiles of the analyzed phenolic compounds. The predominant phenolic compound in each variant was ferulic acid (4-hydroxy-3-methoxycinnamic acid). Its content ranged from 735 µg/g dm (non-fertilized control) to 993 µg/g dm (cattle manure) in basil under optimal conditions, and up to 1559 µg/g dm (HI frass 12.5 g/L) in basil under drought. All fertilizer formulations significantly increased the ferulic acid content compared with the non-fertilized controls. Many physiological activities of ferulic acid have been documented, including antioxidant, antibacterial, anti-inflammatory, anti-thrombotic, and anti-tumor effects [75]. Ferulic acid can be easily absorbed from food and metabolized in the human body, and it has been attributed a health-promoting effect. Chlorogenic and caffeic acids are the second most prominent group of phenolic compounds found in the analyzed basil plants. The use of each of the biofertilizers resulted in a reduction of the content of these phenolic acids, and their content was also influenced by the dose of HI frass applied. *p*-Hydroxybenzoic acid and vanillin were present in relatively lower concentrations. The overall observation is, that the use of biofertilizers significantly influenced not only the antioxidant activity, but also the content of phenolic compounds, which constitute the plant's main response to the changing external conditions.

## Conclusions and outlook

The use of HI frass in basil production is seemingly a very promising avenue towards successful and environmentally friendly fertilizers. Our physiological and biochemical investigations of basil cultivated under drought convergently encourage further exploration of the HI frass in this and other crops. Beyond the benefits reported here, the possible positive impacts at the crops' transcriptomic reactions to drought and other stresses, as well as to the microbiome community and their stress responses, need to be evaluated. The likely discovery of other biofertilizers and of their multifold impacts provides a great leverage to a more sustainable, yet arguably more profitable, agriculture.

## Supporting information

**S1 Table. Means of data from measurements of physiological parameters.** Means and the corresponding one- and two-way ANOVA results, with subsequent Tukey's separation at α =

0.05 are (LSD) are presented. Significance scores are included.
(XLSX)

## Author Contributions

**Conceptualization:** Dominika Radzikowska-Kujawska, Zuzanna Sawinska.

**Data curation:** Marcin Nowicki.

**Formal analysis:** Dominika Radzikowska-Kujawska, Zuzanna Sawinska, Przemysław Łukasz Kowalczewski, Marcin Nowicki.

**Funding acquisition:** Zuzanna Sawinska, Grzegorz Skrzypczak, Marcin Nowicki.

**Investigation:** Dominika Radzikowska-Kujawska, Zuzanna Sawinska, Monika Grzanka, Przemysław Łukasz Kowalczewski, Łukasz Sobiech, Stanisław Świtek, Agnieszka Drożdżyńska, Mariusz Ślachciński.

**Methodology:** Dominika Radzikowska-Kujawska, Przemysław Łukasz Kowalczewski, Mariusz Ślachciński.

**Project administration:** Dominika Radzikowska-Kujawska, Monika Grzanka.

**Resources:** Grzegorz Skrzypczak.

**Supervision:** Dominika Radzikowska-Kujawska.

**Writing – original draft:** Dominika Radzikowska-Kujawska, Przemysław Łukasz Kowalczewski, Marcin Nowicki.

**Writing – review & editing:** Dominika Radzikowska-Kujawska, Przemysław Łukasz Kowalczewski, Marcin Nowicki.

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
