## [Decision Letter · Decision Letter 0]

27 Oct 2022

PONE-D-22-24561Hermetia illucens frass affects the physiological state of basil (Ocimum basilicum L.) and its nutritional value under droughtPLOS ONE

Dear Dr. Nowicki,

Thank you for submitting your manuscript to PLOS ONE. After careful consideration, we feel that it has merit but does not fully meet PLOS ONE’s publication criteria as it currently stands. Therefore, we invite you to submit a revised version of the manuscript that addresses the points raised during the review process.

We look forward to receiving your revised manuscript.

Kind regards,

Umakanta Sarker

Academic Editor

PLOS ONE

Journal Requirements:

3. We note you have included a table to which you do not refer in the text of your manuscript. Please ensure that you refer to Table 3 in your text; if accepted, production will need this reference to link the reader to the Table.

Reviewers' comments:

Reviewer's Responses to Questions

**Comments to the Author**

1. Is the manuscript technically sound, and do the data support the conclusions?

Reviewer #1: Yes

2. Has the statistical analysis been performed appropriately and rigorously? 

Reviewer #1: Yes

3. Have the authors made all data underlying the findings in their manuscript fully available?

Reviewer #1: Yes

4. Is the manuscript presented in an intelligible fashion and written in standard English?

Reviewer #1: No

5. Review Comments to the Author

Reviewer #1: Manuscript Title: Hermetia illucens frass affects the physiological state of basil (Ocimum basilicum L.) and its nutritional value under drought

Comment: The work by Radzikowska et al. advances our understating the biological role for the frass fertilizer for increasing basil physiology and its nutritional status in the presence of insufficient water. The report is well and condensed, as well as technically may be appropriate for PLOS ONE. However, before being able to recommend acceptance, I invite authors to address the following amendments.

1. I made some corrections in the manuscript, please follow those.

2. Extensive editing of English language

3. Cite some references how frass improve plant physiologyand quality.

4. Cite some references how drought affects negatively on physiology and quality of Basil.

5. Materials and Methods should be clearer

6. PLOS authors have the option to publish the peer review history of their article (what does this mean?). If published, this will include your full peer review and any attached files.

Reviewer #1: No

---

## [Author Response · Author response to Decision Letter 0]

9 Nov 2022

Dear Dr. Umakanta Sarker,

Thank you for your Editorial service towards the peer-review of our manuscript. The Authors are very grateful to the Reviewer for pointing out deficiencies to make this a stronger submission. We added the necessary information. We hope that the revised manuscript now meets the requirements for publication in PLoS One.

Respectfully yours,

Marcin Nowicki 

#Rewiever 1

Comment: The work by Radzikowska et al. advances our understating the biological role for the frass fertilizer for increasing basil physiology and its nutritional status in the presence of insufficient water. The report is well and condensed, as well as technically may be appropriate for PLOS ONE. However, before being able to recommend acceptance, I invite authors to address the following amendments.

Response: Thank you for your helpful comments and suggestions for improvement; these made our report stronger.

Comment: 1. I made some corrections in the manuscript, please follow those.

Response: Thank you for your work. We genuinely appreciate it. 

Comment: 2. Extensive editing of English language

Response: We carefully scanned and edited the Manuscript and accompanying files throughout, to the best of our collective command of English. We noted the often-missing spaces, likely introduced during the conversion to .pdf by the Submission system. We could not address the generic character of that comment without any specific passages needing “extensive editing” in the Reviewer’s opinion.

Comment: 3. Cite some references how frass improve plant physiology and quality.

Response: It has been added to the MS (69-80).

“Use of Hermetia illucens frass (HI frass) as an additive to the growing media in the soilless production of basil, lettuce, and tomato increased the yield (drought mass) [12]. Also, the study of the compost produced as a result of the bioconversion of coffee husks by the larvae of the H. illucens in the cultivation of lettuce showed a positive effect of such fertilizers on plant height, number of leaves, leaf area, and chlorophyll content [13]. Recently published data indicate that the use of insect frass as a soil component in plant cultivation provides nitrogen and other nutrients, which consequently increases plant biomass and nutrient content [11,14]. Quality of the HI frass is high in terms of N, P, and K levels [15]. Use of HI frass from larvae increased the soil organic matter and the residual nutrients content, as well as the enzymatic activity of dehydrogenase and β-glucosidase [16].

Comment: 4. Cite some references how drought affects negatively on physiology and quality of Basil.

Response: It has been added to the MS (99-111).

“Basil is an annual plant that can be successfully grown in warm and sunny conditions [30]. The optimal growth temperature ranges from 298 to 303 K [31]. Basil, however, is quite sensitive to drought stress [32]. Drought stress triggers a number of various physiological responses that have negative effects on plant growth and development. Plants are adapted to the short-term effect of the stress factor, whereas under prolonged drought stress, the rate of water absorption from the leaves is increased, which consequently reduces their hydration. This can result in closure of the stomata and in reduced cell enlargement and growth [33]. In addition, drought also causes a reduction in the activity of the electron transport chain, which leads to the accumulation of reactive oxygen species (ROS) that are toxic at elevated levels [34]. ROS may damage nucleic acids, proteins, photosynthetic pigments, and membrane lipids [34,35], and that damage may impair plant development, yield, and even lead to plant death.

Comment: 5. Materials and Methods should be clearer

Response: The description of the methods is based on the literature data and manufacturers' recommendations. We agree with the Reviewer that it could be clearer as it is very detailed and extensive. But, with the reviewer's consent, we would like to leave that section as currently described. Our accurate descriptions of the analytical methods allow their independent reproduction.

Comment: What was the water holding capacity of the experimental soil at 80% Field capcity and what was the soil pH?

Response: Soil pH was 6.2; this information has been added to the text (124). The water capacity was of lesser import for us, because the only differing factor throughout the experiment was the fertilization regime. Please note, data for full parallel characteristics under control (non-stressed; regularly watered) and drought (stressed) conditions are presented; these were generated in the same soil throughout.

Comment: ℃

Response: It means Celsius degree. It has been calculated to K, according to SI units.

Comment: Where is figure?

Response: The figure is attached as a separate file. It was successfully produced into the .pdf for us.

---

## [Decision Letter · Decision Letter 1]

20 Dec 2022

Hermetia illucens frass improves the physiological state of basil (Ocimum basilicum L.) and its nutritional value under drought

PONE-D-22-24561R1

Dear Dr. Nowicki,

We’re pleased to inform you that your manuscript has been judged scientifically suitable for publication and will be formally accepted for publication once it meets all outstanding technical requirements.

Kind regards,

Umakanta Sarker

Academic Editor

PLOS ONE

Additional Editor Comments (optional):

Reviewers' comments:

Reviewer's Responses to Questions

**Comments to the Author**

1. If the authors have adequately addressed your comments raised in a previous round of review and you feel that this manuscript is now acceptable for publication, you may indicate that here to bypass the “Comments to the Author” section, enter your conflict of interest statement in the “Confidential to Editor” section, and submit your "Accept" recommendation.

Reviewer #1: All comments have been addressed

2. Is the manuscript technically sound, and do the data support the conclusions?

Reviewer #1: Yes

3. Has the statistical analysis been performed appropriately and rigorously? 

Reviewer #1: Yes

4. Have the authors made all data underlying the findings in their manuscript fully available?

Reviewer #1: Yes

5. Is the manuscript presented in an intelligible fashion and written in standard English?

Reviewer #1: Yes

6. Review Comments to the Author

Reviewer #1: This is a good peace of work which will contribute to the future research works in this aspect. Author has responses all quarries properly. So the manuscript may be considered for publication.

7. PLOS authors have the option to publish the peer review history of their article (what does this mean?). If published, this will include your full peer review and any attached files.

Reviewer #1: No

---

## [Editor Report · Acceptance letter]

23 Dec 2022

PONE-D-22-24561R1 

*Hermetia illucens* frass improves the physiological state of basil (*Ocimum basilicum* L.) and its nutritional value under drought 

Dear Dr. Nowicki:

I'm pleased to inform you that your manuscript has been deemed suitable for publication in PLOS ONE. Congratulations! Your manuscript is now with our production department. 

Kind regards, 

on behalf of

Professor Umakanta Sarker 

Academic Editor

PLOS ONE